# Select Porcine Elongation Factor 1α Sequences Mediate Stable High-Level and Upregulated Expression of Heterologous Genes in Porcine Cells in Response to Primate Serum

**DOI:** 10.3390/genes12071046

**Published:** 2021-07-07

**Authors:** Wu-Sheng Sun, Hyeon Yang, Jin Gu No, Haesun Lee, Nahyun Lee, Minguk Lee, Man-Jong Kang, Keon Bong Oh

**Affiliations:** 1Animal Biotechnology Division, National Institute of Animal Science, Rural Development Administration, Wanju-gun 55365, Korea; sunwsh@korea.kr (W.-S.S.); yh0415@korea.kr (H.Y.); shrkftm@gmail.com (J.G.N.); leehs1498@korea.kr (H.L.); dngkgus@naver.com (N.L.); bosuk7842@korea.kr (M.L.); 2Department of Animal Science, Chonnam National University, Gwangju 61186, Korea; mjkang@jnu.ac.kr

**Keywords:** promoter, porcine elongation factor 1α, expression, porcine cells, xenogeneic serum

## Abstract

Genetically engineered (GE) pigs with various combinations of genetic profiles have been developed using heterologous promoters. This study aimed to identify autologous promoters for high and ubiquitous expression of xenotransplantation relevant genes in GE pigs. A 1.4 kb upstream regulatory sequence of porcine elongation factor 1α (pEF1α) gene was selected and isolated for use as a promoter. Activity of the pEF1α promoter was subsequently compared with that of the cytomegalovirus (CMV) promoter, CMV enhancer/chicken β-actin (CAG) promoter, and human EF1α (hEF1α) promoter in different types of pig-derived cells. Comparative analysis of luciferase and mutant human leukocyte antigen class E-F2A-β-2 microglobulin (HLA-E) expression driven by pEF1α, CMV, CAG, and hEF1α promoters revealed the pEF1α promoter mediated comparable expression levels with those of the CAG promoter in porcine ear skin fibroblasts (PEFs) and porcine kidney-15 (PK-15) cells, but lower than those of the CAG promoter in porcine aortic endothelial cells (PAECs). The pEF1α promoter provided long-term stable HLA-E expression in PEFs, but the CAG promoter failed to sustain those levels of expression. For xenogeneic serum-induced cytotoxicity assays, the cells were cultured for several hours in growth medium supplemented with primate serum. Notably, the pEF1α promoter induced significant increases in luciferase and HLA-E expression in response to primate serum in PAECs compared with those driven by the CAG promoter, suggesting the pEF1α promoter could regulate temporal expression of heterologous genes under xenogeneic-cytotoxic conditions. These results suggest the pEF1α promoter may be valuable for development of GE pigs spatiotemporally and stably expressing immunomodulatory genes for xenotransplantation.

## 1. Introduction

Genetically engineered (GE) pigs for use in xenotransplantation have been considered an alternative donor source to provide a therapeutic resource for patients with end-stage organ failure as pigs share many similarities with humans regarding anatomy, physiology, and phylogeny [1]. During the process of producing GE pigs, promoter selection and optimization to drive and regulate the spatial and temporal expression of target genes relative to their intended purpose is a critical step. Currently, both viral and mammalian gene promoters have been used in GE pigs in effort to express immunomodulatory and/or coagulation-modulatory proteins for xenotransplantation.

Cytomegalovirus (CMV) enhancer/chicken β-actin (CAG) promoter has been predominately used for ubiquitous and constitutively high-level expression of target genes in GE pigs [2,3,4]. However, accumulating evidence suggests this viral promoter is vulnerable over time to methylation-mediated transcriptional silencing due to host-cell epigenetic defense mechanisms that have emerged [5,6]. Mouse H-2Kb, human α-globin, and human eukaryotic translation elongation factor 1α (*hEF1α*) promoters have also been used for generation of GE pigs; however, expression levels of target genes using these promoters are unequable [7]. Alternatively, full or partial genomic sequences of immunomodulatory and anticoagulation genes, including autologous promoters, have been successfully adopted in order for GE pig to possess expression specificity for particular genes [7,8]. Nevertheless, the availability of promoters to construct heterologous gene-expression cassettes may be limited with lengths exceeding 4 kb [7,9]. Therefore, it is necessary to explore promoters with high activity and small size.

Inducible promoters allow target-gene expression to be upregulated under endogenous and exogenous stimuli, making it possible to more precisely regulate temporal expression of genes that may be harmful to the development of GE animals [10]. Indeed, the development of cytokine inducible promoters, complement factor 3, serum amyloid A3, and acute-phase protein for regulating reporter gene expression in response to lipopolysaccharide in vivo have been reported [11]. Recently, Fischer et al. [12] demonstrated that optimized human C-C motif chemokine ligand 2 and tumor necrosis factor α induced protein 3, and porcine A20 promoters are able to direct high-level expression of the developmental defect gene *LEA29Y* in response to TNFα and IL1ß, which are well known pro-inflammatory cytokines that are initially released as part of xenograft rejection mechanisms [13,14]. The development of these promoters is of great significance to the production of GE pigs that can accurately express immunomodulatory genes for xenotransplantation.

In the current study, we identified for the first time a novel function of a 1.4 kb sequence region of porcine elongation factor 1α (*pEF1α*) gene. We characterized this sequence in vitro as a xenogeneic serum-inducible promoter. The *pEF1α* promoter supported heterologous gene expression levels comparable with that of the CAG promoter in primary pig ear skin fibroblasts (PEFs) and porcine kidney-15 (PK-15) cells under basic culture conditions. However, the *pEF1α* promoter drove increased expression levels of heterologous genes in porcine aortic endothelial cells (PAECs) compared to that of the CAG promoter when induced with xenogeneic serum. Notably, we were able to confirm that the *pEF1α* promoter activity was sustained at the initial level during long-term culture, whereas the CAG promoter activity was downregulated with time. We suggest that the *pEF1α* promoter of this study may be useful for generating GE pigs by precisely directing the spatial and temporal expression of single and multiple genes in order to attenuate the immune response during xenotransplantation.

## 2. Materials and Methods

### 2.1. Promoter and Expression Vector Construction

The CMV, CAG, and *hEF1α* promoters were isolated from the pcDNA^TM^ 3.1 (+) vector (Invitrogen, CA, USA), pCAGMKOSiE vector (Addgene, MA, USA), and pBudCE4.1 vector (Invitrogen), respectively. We searched the National Center for Biotechnology Information (NCBI) Sscrofa11.1 database using the *hEF1α* sequences from the human genome GRCh38 assembly, chr6: 73,521,970–73,518,347 as a template and identified a conserved region of *pEF1α* sequence (chr1: 92,421,084–92,424,225), which spanned six exons and approximate 1.4 kb of upstream regulatory sequence (URS) (Figure 1A). The selected *pEF1α* promoter and an expression cassette including human leukocyte antigen class E (HLA-E) complementary DNA (cDNA) with a mutation in the leader peptide region (HLA-E Cw0304) connected with β-2 microglobulin (B2M) cDNA [15] were synthesized by Thermo Fisher Scientific (Waltham, MA, USA). The cassette was subsequently inserted to the pcDNA3.1^TM^ (+) vector and the original vector promoter was replaced with the *pEF1α* or CAG promoter. Comparative analysis of the *pEF1α* and *hEF1α* promoter sequences was performed using DNAMAN software (Lynnon Biosoft, Quebec, QC, Canada) and the mVISTA program in LAGAN mode with default parameters [16].

### 2.2. Cell Culture, Transfection, Selection, Serum Induction, and Genotyping

PEFs and PAECs were isolated from one-month-old α1,3-galactosyltransferase knockout (GTKO) pigs and cultured as previously described [17]. PK-15 cells were maintained in Dulbecco’s Modified Eagle’s Medium (DMEM; Invitrogen) supplemented with 10% (*v/v*) fetal bovine serum (FBS) and 1× Antibiotic-Antimycotic (Thermo Fisher Scientific). The methods for transfecting the PEFs and PAECs were described previously [17]. The PK-15 cells were transfected using a Lipofectamine 3000 Kit (Invitrogen) according to manufacturer’s protocol. A pMAX GFP plasmid (Lonza, Basel, Switzerland) was co-transfected with the experimental constructs to help normalize the transfection efficiency. The transfected PEFs were selected using 600 μg/mL neomycin (Thermo Fisher Scientific) for 10 days (D0). The cells were subsequently maintained in selection medium for an additional 4 days (D4), 8 days (D8), 16 day (D16), or 20 days (D20). Serum induction of the PAECs was performed using 15% primate serum (Xenia Inc., Seong-nam, Korea). Porcine serum obtained from whole blood of a 7-month-old male GTKO pig bred at National Institute of Animal Science (Wanju-gun, JB, Korea) was used as a control. Genomic DNA of transfected and neomycin-selected cells was extracted using a DNeasy Blood and Tissue Kit (Qiagen, Hilden, Germany). Polymerase chain reaction (PCR) analysis was performed using Prime Taq Premix (Genet Bio, Nonsan, Korea) with primers presented in Table 1.

### 2.3. Luciferase Assay

Four different promoters were individually subcloned into a pGL3-Basic Luciferase Reporter Vector (Promega, Madison, WI, USA) and then transfected into PEFs, PK-15 cells, and PAECs. At 48 h post transfection, the cells were washed with Hank’s Balanced Salt Solution (HBSS, Invitrogen) and then analyzed using a Dual-luciferase^®®^ Reporter Assay System (Promega), according to the manufacturer’s instructions. Luminescence from each sample was measured using a Centro LB 960 luminometer (Berthold, Bad Wildbad, Germany). All values are normalized relative to values for the CAG promoter.

### 2.4. Real-Time Quantitative PCR

Total RNA was extracted from the cells using a RNeasy Mini Kit (Qiagen). The cDNAs were synthesized using a Superscript IV First-Strand Synthesis Kit (Invitrogen). Real-time quantitative PCR (qPCR) was performed in triplicate using Power SYBR Green PCR master mix (Applied Biosystems, Foster City, CA, USA) and a StepOne Real-Time PCR System (Applied Biosystems). Sequences of the primers used are presented in Table 1.

### 2.5. Western Blot Analysis

Whole cell proteins were isolated using mammalian protein extraction reagent M-PER (Invitrogen) supplemented with a protease inhibitor cocktail (Thermo Fisher Scientific). Protein concentrations were determined using the Bradford assay (Bio-Rad, CA, USA). Equal amounts of protein samples (10 μg) were loaded onto NuPAGE™ 4–12% SDS-PAGE gels (Invitrogen), electrophoresed, and transferred to polyvinylidene fluoride membranes (Invitrogen). The membranes were subsequently incubated with primary antibodies overnight at 4 °C followed by secondary antibodies for 1 h at room temperature. Mouse anti-HLA-E monoclonal antibody (Acris, Rockville, MD, USA), mouse anti-B2M, or anti-β-actin antibody (Santa Cruz, TX, USA) were used as primary antibodies and goat anti-mouse IgG-horseradish peroxidase (HRP) antibody (Santa Cruz) was used as the secondary antibody. The immunostained blots were visualized using Amersham ECL Prime Western Blotting Detection Reagent (GE Healthcare, Chalfont St Giles, Buckinghamshire, UK) and the signal detected using an EZ-Capture II chemiluninescence charge-coupled device (CCD) imaging system (ATTO, Amherst, NY, USA).

### 2.6. Flow Cytometry

The transfected neomycin-selected PEFs were harvested and incubated with phycoerythrin (PE)-conjugated anti-human HLA-E antibody (Milteny Biotec, Bergish Gladbach, Germany) in staining buffer consisting of 1× phosphate-buffered saline supplemented with 0.5% (*w/v*) bovine serum albumin (BSA). The cells were washed three times with staining buffer and analyzed by flow cytometry using a FACS CaliburTM (BD Bioscience, NJ, USA). A PE-conjugated mouse IgG1κ antibody (Milteny Biotec) was used as an isotype control.

### 2.7. Statistical Analysis

All experimental results underwent statistical analysis using GraphPad Prism 5.03 software (GraphPad Software Inc., San Diego, CA, USA). Statistical significance for the promoter screening reporter assays and endogenous mRNA expression of the HLA-E gene were assessed using one-way analysis of variance (ANOVA). Two-way ANOVA was implemented for analysis of the xenogeneic study results. All the data are represented in mean ± standard error of the mean (SEM). A *p*-value less than 0.05 was considered statistically significant.

## 3. Results

### 3.1. Sequence Analysis of the pEF1α Upstream Region

Comparative analysis of the selected upstream sequences of *pEF1α* and *hEF1α* revealed that the porcine sequence shared 65.76% homology with that of its human counterpart (Figure 1A) and the distribution pattern of CpGs differed slightly between the porcine and human sequences (Figure 1B). However, conserved CpG islands (Figure 1B) and several putative transcription-factor binding sites, such as those for Sp1, GATA, NF-κB, and C/EBP, were identified within the upstream region of both the *hEF1α* and *p**EF1α* (Appendix A). We selected a 1421 bp-length segment in the upstream region of *pEF1α* (chr1: 92,421,241–92,422,700), which included predicted CpG islands at the 5’ end (Figure 1B).

### 3.2. The pEF1α Promoter Led to High-Level Heterologous Gene Expression in Porcine Cells

Four luciferase reporter vectors (Figure 2A) were individually transiently transfected into PEFs, PK-15 cells, and PAECs. The results of luciferase assay revealed that the *pEF1α* promoter led to a 2.2-fold higher level of luciferase expression than that of the CAG promoter in PEFs, which was a statistically significant difference (Figure 2B). No significant difference was observed in the levels of expression between the *pEF1α* and CAG promoters in PK-15 cells (Figure 2C). However, the level of luciferase expression driven by the *pEF1α* promoter was significantly lower compared with that of the CAG promoter in PAECs (Figure 2D). Interestingly, activity of the CMV and *hEF1α* promoters, which are frequently used for mammalian gene expression in vitro and in vivo, were shown to be significantly less than that of the CAG promoter activity in the porcine cells tested (Figure 2B–D).

To determine whether the *pEF1α* promoter regulated heterologous gene expression independent of the gene linked, we constructed HLA-E expression cassettes driven by *pEF1α* and CAG promoters (Figure 3A) and introduced the cassettes into PEFs, PK-15 cells, and PAECs. The results indicate that the expression levels of HLA-E mRNA (Figure 3B) and protein (Figure 3C) were consistent with that of luciferase expression (Figure 2); significantly higher in PEFs, comparable in PK-15 cells, and significantly lower in PAECs when driven by the *pEF1α* promoter. Taken together, these results indicate that the *pEF1α* promoter activity was comparable with that of the CAG promoter regarding the regulation if gene expression in porcine cells, except for PAECs in which expression was lower.

### 3.3. The pEF1α Promoter Induced Heterologous Gene Expression in Response to Xenogeneic Serum

The PAECs transiently transfected with luciferase (Figure 2A) and HLA-E (Figure 3A) expression cassettes were cultured for 7 h in endothelial culture medium supplemented with 15% normal primate serum or porcine serum. Porcine serum treatment did not affect the *pEF1α* promoter activity or CAG promoter activity with respect to the expression of luciferase or the HLA-E transgene (Figure 4A) compared to that of the results for basic endothelial culture conditions with 5% fetal bovine serum (Figure 2D and Figure 3B). Interestingly, treatment with primate serum resulted in significantly higher expression of luciferase and HLA-E under the control of the *pEF1α* promoter compared to that under control of the CAG promoter (Figure 4B). These results suggest the *pEF1α* promoter was able to induce increased expression of the target gene under conditions similar to xenogeneic stress, at least in PAECs. We were not able to detect a specific increase of endogenous *pEF1α* expression following primate serum treatment (Figure 4C). Notably, tissue factor (*TF*), which has been identified as a gene upregulated during cardiac xenograft rejection [18], exhibited serially increased expression depending on time of primate serum treatment (Figure 4D). We were able to confirm that intercellular adhesion molecule 2 (*ICAM2*), a gene specifically expressed in endothelium, was consistently expressed (Figure 4E), indicating the upregulation of endogenous *TF* expression and exogenous *pEF1α*-luciferase and *pEF1α*-HLA-E expression were a consequence of promoter induction in response to primate serum. We suggest that the *pEF1α* sequences selected in this study lost regulatory elements essential for consistent expression of *pEF1α*, but obtained novel function that was inducible by primate serum.

### 3.4. The pEF1α Promoter, but Not the CAG Promoter, Retained High-Level HLA-E Expression during Long-Term Culture

PEFs transfected for HLA-E expression under *pEF1α* and CAG promoters were selected and serially harvested, as described in the materials and methods section. Genotype analysis showed that the selection procedure was successful (Figure 5A) without causing morphological changes in the cells (Appendix A). The qPCR analysis showed the *pEF1α* promoter led to consistent expression of HLA-E, but the CAG promoter led to a dramatic downregulation of HLA-E expression starting from D8 compared with that at D4 (Figure 5B). The qPCR results were confirmed by western blot analysis, which demonstrated similar patterns (Figure 5C). Furthermore, flow cytometry analysis clearly showed that the *pEF1α* promoter retained expression levels over time; this contrasted with that of the CAG promoter, which showed a decreased expression level by D20 (Figure 5D). Therefore, we demonstrated the *pEF1α* promoter, in contrast to that of the CAG promoter, possessed activity that allowed it to consistently control target gene expression comparable to the levels initially expressed.

## 4. Discussion

GE pigs that ablate or attenuate genes involved in pig-to-primate rejection mechanisms are considered a key goal for successful xenotransplantation. Currently, multiple GE pigs have been generated, including those with immunomodulatory, anti-coagulation, anti-inflammatory, and anti-apoptotic genetic changes in a GTKO background. The choice of promoters among the various studies on GE pig production differs depending on the purpose of the target gene and whether ubiquitous and/or specific expression is desired. Promoters for ubiquitous expression seem to be suitable for expressing human complement regulating genes and immune-cell response factor genes, such as CD46, CD55, human HLAs, and CD47. The *ICAM2* promoter has been frequently used for specific expression of anti-coagulation and anti-inflammatory proteins, including tissue factor pathway inhibitor (TFPI), thrombomodulin (THBD), C1 inhibitor, and A20 [3,19,20]. In previous studies, we reported the production of GTKO pigs carrying human CD46 (GTKO/CD46) or THBD (GTKO/CD46/THBD) driven by the CMV promoter [21,22], CAG promoters for ubiquitous expression, and the pig *ICAM2* promoter for endothelium-specific expression of THBD and CD73 [23]. We demonstrated that the CMV promoter was moderately active for CD46 expression in GTKO/CD46 pigs, and the CAG promoter was rather strong for CD46 expression in GTKO/CD46/THBD pigs. In the current study, we successfully developed the *pEF1α* promoter, an autologous gene promoter of pigs. This promoter ubiquitously regulated high-level expression of target genes compared to that of the CAG promoter and eliminated potentially unstable expression, both improvements over the CAG promoter for the generation of GE pigs. Furthermore, we were able to confirm that the *pEF1α* promoter allowed temporal gene expression induced by xenogeneic stress conditions.

The CAG promoter showed consistent high-level expression of luciferase compared to that of the CMV, *hEF1α*, and *pEF1α* promoters, independent of the cell type. Consistent with this, the CAG promoter has been reported in previous studies as a valuable promoter for GE pigs. Considering expression levels of heterologous genes, the CAG promoter may be the best option for use in the generation of GE pigs for xenotransplantation. However, the CAG promoter failed in the current study to drive stable expression of HLA-E in porcine cells, consequently raising concern that the level of HLA-E expression in CAG-HLA-E pigs may be different between generations of pigs and/or among individual pigs. To develop a novel autologous gene promoter for pigs that essentially provides stable expression of the target gene at high levels, we isolated and analyzed the activity of porcine *CD46*, *ICAM2*, *β-ACTIN*, and *pEF1α* promoters. The results were the *pEF1α* promoter was comparable to that of the CAG promoter with respect to the expression levels. Moreover, this study showed that the *pEF1α* promoter regulated the stable expression of target genes at initial levels and did so long-term, clearly demonstrating that the *pEF1α* promoter is a reasonable choice for use in GE pig production.

We also identified several conserved CpG islands and putative transcription-factor binding sites in the current study, such as GATA, stress-responsive HSF [24], inflammatory-responsive NF-κB, and C/EBP motifs of the *pEF1α* promoter [25]. These transcription-factor binding sites were shown in the *hEF1α* promoter and are well known to enhance gene expression. Wang et al. reported that the *hEF1α* promoter increases gene expression compared with that of the CMV and CAG promoters in CHO cells [26]. Among the four different promoters tested in our current study, activity of the *pEF1α* promoter was highest and that of the *hEF1α* promoter was lowest in PEFs and PAECs (Figure 2). This could have been attributable to the *hEF1α* promoter being suppressive in porcine cells. We suggest that the *hEF1α* promoter would be a poor choice for GE pig production when it comes to xenotransplantation.

We recently reported a list of differentially expressed genes in xenografted hearts of GTKO pigs into primates [18]. The list included a significant increase in *TF* expression, but no change in expression levels of *EF1α* or *ICAM 2*. The genes exhibited a similar pattern in endothelial cells cultured with medium supplemented with primate serum, suggesting that the conditions of culture with primate serum are likely to at least partially mimic the rejection mechanisms in cardiac xenotransplantation (Figure 4). Therefore, we evaluated the effect of primate serum stimulation on the activity of the *pEF1α* and CAG promoters. The results demonstrated that the *pEF1α* promoter responded with significantly increased levels of luciferase and HLA-E expression compared with that of the CAG promoter. The precise induction mechanism(s) of the *pEF1α* promoter in response to primate serum remains unknown. Regardless of the mechanism, the novel function of the *pEF1α* promoter with a swift and dynamic upregulation of gene expression in response to xenogeneic stimuli should prove beneficial for investigating xenotransplantation by dramatically upregulating target immune regulatory genes involved during mechanisms of graft rejection.

## 5. Conclusions

We developed an autologous promoter to provide a high-level expression of target genes during the production of GM pigs, independent of the specific generation or individual pig. Specifically, we isolated the *pEF1α* promoter and analyzed its consistency and comparative expression level relative with that of the CAG promoter in porcine cells, including PEFs, PK-15 cells, and PAECs. The results demonstrated that the *pEF1α* promoter was able to mediate stable consistent expression of heterologous target genes at a high level in PEFs, while the CAG promoter led to a dramatic downregulation of target gene expression. Notably, the *pEF1α* promoter showed a novel function in responding to xenogeneic serum, resulting in a significant increase of target gene expression in PAECs. Taken together, we believe the *pEF1α* promoter may be valuable for the development of GM pigs spatiotemporally and stably expressing immunomodulatory genes for xenotransplantation.

## Figures and Tables

**Figure 1 genes-12-01046-f001:**
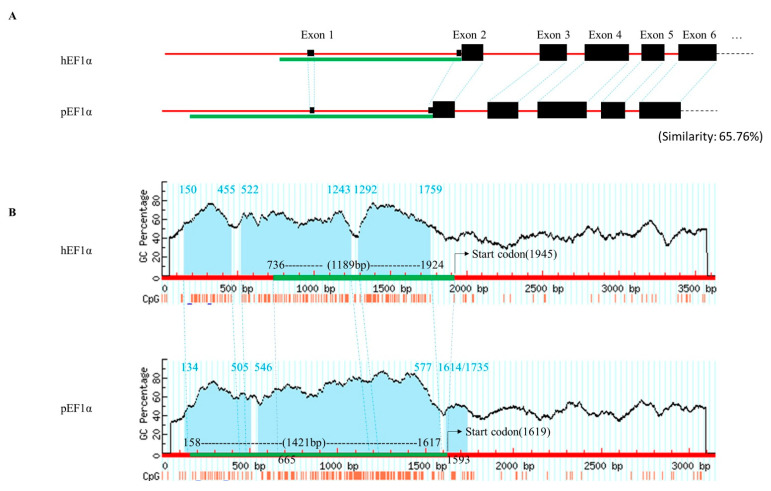
Sequence alignment of *hEF1α* and *pEF1α*, including the URS. (**A**) Structure comparison of the selected URS (red lines) of *hEF1α* and *pEF1α*. Exons are shown in black boxes. (**B**) Pairwise alignment of the URS of *hEF1α* and *pEF1α*. Three predicted CpG islands are filled with blue and indicated as numbers on top of the graph. Location of the CpG dinucleotides is indicated by vertical red bars under the x-axis. Ortholog gene sequences of *pEF1α* and *hEF1α* are connected on the map with dotted blue lines. Green lines indicate the locations of the promoters used in the current study. *hEF1α*, human elongation factor 1α; *pEF1α*, porcine elongation factor 1α; URS, upstream regulatory sequence.

**Figure 2 genes-12-01046-f002:**
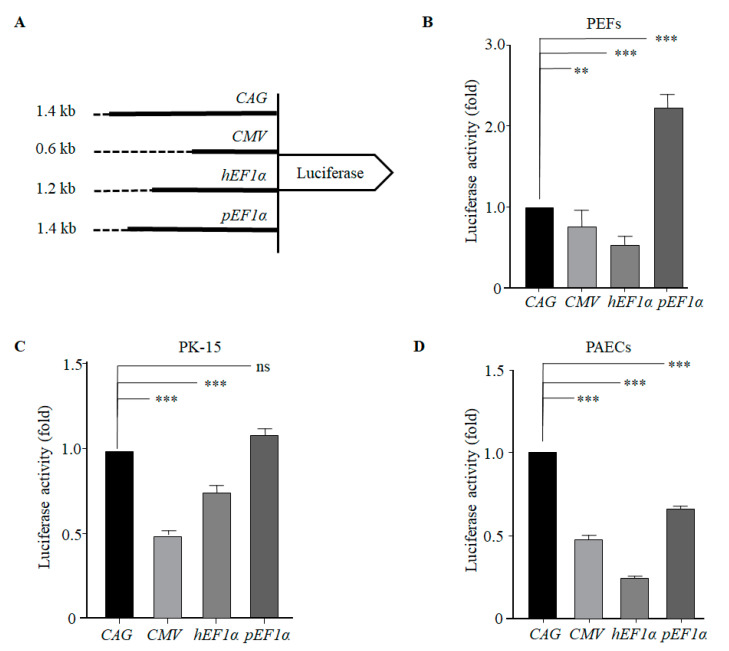
Relative activity of various promoters in regulating luciferase expression in different types of porcine cells. (**A**) Schematic diagram of the luciferase reporter constructs used in the study. Numbers on the left indicate the approximate length of the respective promoters. (**B**) Promoter activity in PEFs. (**C**) Promoter activity in PK-15 cells. (**D**) Promoter activity in PAECs. All values are normalized relative to values for the CAG promoter. CAG, cytomegalovirus enhancer/chicken β-actin promoter; CMV, cytomegalovirus promoter; *hEF1α*, human elongation factor 1α; *pEF1α*, porcine elongation factor 1α; PEFs, porcine ear skin fibroblasts; PAECs, porcine aortic endothelial cells. Values are means ± SEM calculated from three replicates; ns, no significant difference; **, *p* < 0.01; ***, *p* < 0.001.

**Figure 3 genes-12-01046-f003:**
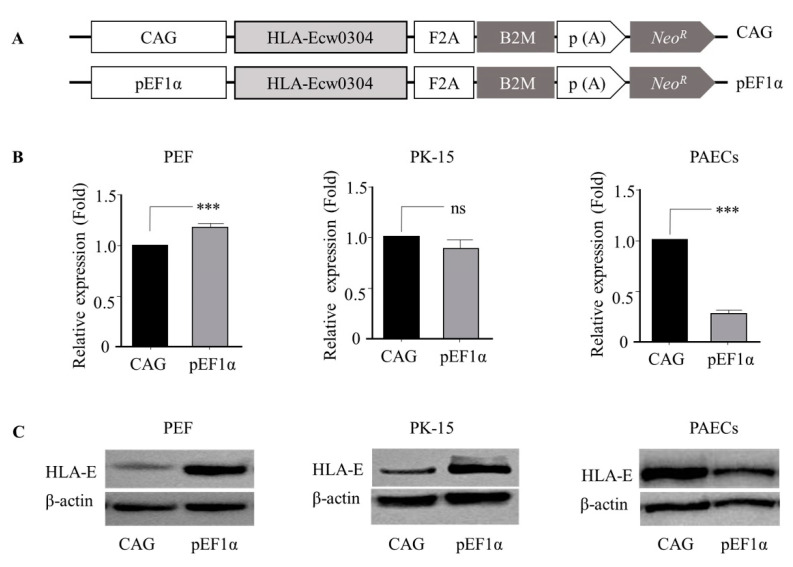
Comparative activity of the *pEF1α* promoter with that of the CAG promoter in regulating an immunomodulatory gene in different types of porcine cells. (**A**) Schematic of the cassettes for HLA-Ecw0304 and B2M (HLA-E) co-expression. Comparative expression of mRNA levels (**B**) and protein levels (**C**) of HLA-E in PEFs, -15 cells, and PAECs. *pEF1α*, porcine elongation factor 1α; HLA-E, human leukocyte antigen E; F2A, 2A self-cleaving peptides; B2M, β-2 microglobulin; p(A), poly A; Neo^®®^, neomycin resistance. PEFs, porcine ear skin fibroblasts; PAECs, porcine aortic endothelial cells. ***, *p* < 0.001.

**Figure 4 genes-12-01046-f004:**
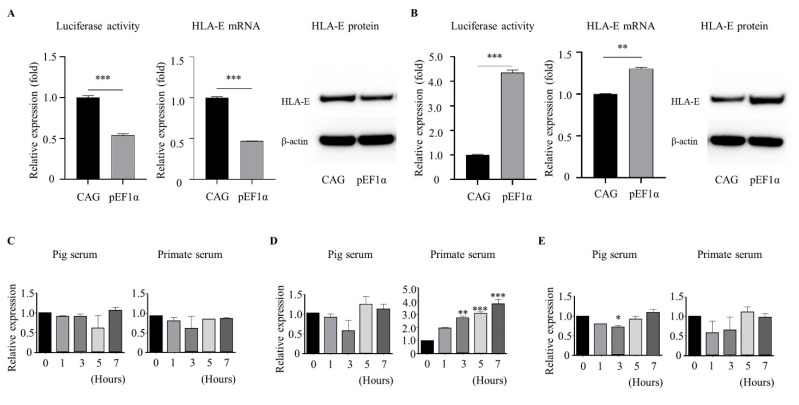
The *pEF1α* promoter mediates significant increases in heterologous gene expression in response to primate serum in PAECs. (**A**) Luciferase activity, mRNA expression levels, and protein expression levels of HLA-E transgenes in PAECs transiently transfected with the expression cassettes presented in Figure 3A, followed by incubation in culture medium supplemented with pig serum. (**B**) Luciferase activity, mRNA expression levels, and protein expression levels of HLA-E transgenes in transfected PAECs incubated in culture medium supplemented with primate serum. (**C**–**E**) Expression pattern of endogenous *pEF1α* (**C**), *TF* (**D**), and *ICAM2* (**E**) in PAECs cultured with porcine serum and primate serum for 1, 2, 3, 5, and 7 h. CAG, cytomegalovirus enhancer/chicken β-actin promoter; *pEF1α*, porcine elongation factor 1α; HLA, human leukocyte antigen E; *ICAM2*, intercellular adhesion molecule 2; *TF*, tissue factor. Values are means ± SEM calculated from three replicates. *, *p* < 0.05; **, *p* < 0.01; ***, *p* < 0.001.

**Figure 5 genes-12-01046-f005:**
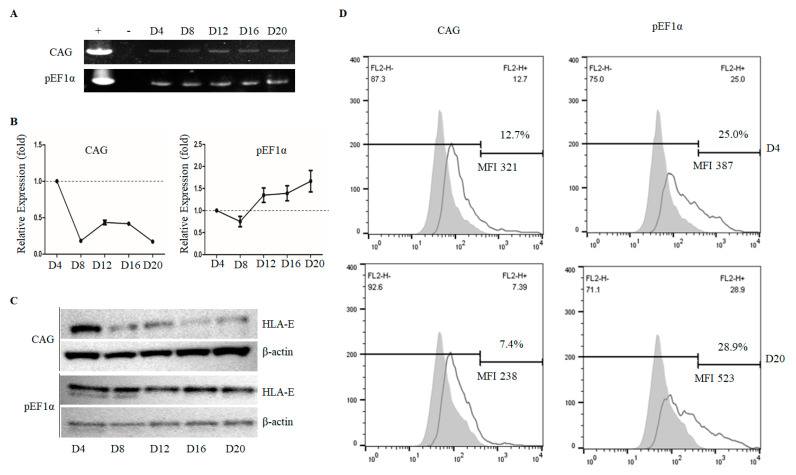
The *pEF1α* promoter mediates stable HLA-E expression during long-term culture. (**A**) CAG- and the *pEF1α*-HLA-E expression cassettes were transfected into PEFs and then selected as described in detail in the Materials and Methods. Genotype analysis revealed that the transfectants carried the cassettes on day 4 (D4), day 8 (D8), day 12 (D12), day 16 (D16), and day 20 (D20) after 10 days of primary selection. (**B**) Comparative analysis of HLA-E mRNA expression at D20 compared with that at D4. Values are means ±SEM calculated from three replicates. (**C**) Western blot analysis of HLA-E expression on different days of selection. (**D**) Flow cytometry analysis for HLA-E at D20 compared with that at D4. “+”, positive control; “−”, negative control; CAG, cytomegalovirus enhancer/chicken β-actin promoter; *pEF1α*, porcine elongation factor 1α; HLA, human leukocyte antigen E; MFI, mean fluorescence intensity.

**Table 1 genes-12-01046-t001:** Primers used in this study.

Experiment	Gene Symbol	Amplicon Size (bp)	Primer (5′→3′)
Real-time PCR	hHLA-E	501	GGGCTACCCGAGCCCGTCACCCTGAGATGG
			TTCAATTCTCTCTCCATTCTTCAGTAAGTC
	pGAPDH	219	TCGGAGTGAACGGATTTG
			CCTGGAAGATGGTGATGG
	pTF	181	AACTGAATGTGACCGTAGAAGCTG
			CCTTTATCCACGTCAATCAGAAAC
	pEF1α	136	TGGATTGCATTCTACCACCA
			ACCATGCCAGGTTTGAGAAC
	pICAM2	117	CGTGCTGCTCTTCTTGTTTG
			CACCTCAGCCTCCTCTGAAC
Genotyping	hHLA-E	1274	ATTTCCACACTTCCGTGTCC
			TTCAATTCTCTCTCCATTCTTCAGTAAGTC

GAPDH, glyceraldehyde 3-phosphate dehydrogenase; HLA-E, human leukocyte antigen E; *TF*, tissue factor; *EF1α*, elongation factor 1α; *ICAM2*, intercellular adhesion molecule 2; h, human; p, pig.

## Data Availability

Data is contained within the article or Appendix A.

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
