# Peer review of "Select Porcine Elongation Factor 1α Sequences Mediate Stable High-Level and Upregulated Expression of Heterologous Genes in Porcine Cells in Response to Primate Serum"

_genes, 2021, doi:10.3390/genes12071046_

Round 1

Reviewer 1 Report

The aim of the study should be stated more precisely. In the Abstract aim is vague.

In the Introduction the authors wrote what they did, but not what for. Besides, the Introduction is too succinct, not fully enlighten the studied topic and may be plotted more interesting.

There is no actual discussion in the Discussion section. The authors majorly cited their own previous results. 

Author Response

→ We appreciate the time and efforts by the reviewer in reviewing this manuscript. We have addressed all issues indicated in the review report and showed in blue. We hope that the revised version can meet the journal publication requirements.

  • The aim of the study should be stated more precisely. In the Abstract aim is vague.

→ As suggested by the reviewer, we updated the abstract (line 13~14).

  • In the Introduction the authors wrote what they did, but not what for. Besides, the Introduction is too succinct, not fully enlighten the studied topic and may be plotted more interesting.

→ Thank you very much for your opinion. We have summarized the introduction at the end of each paragraph (line 54~55, line 66~67).

  • There is no actual discussion in the Discussion section. The authors majorly cited their own previous results. 

→ This study is conducted on the basis of our previous results, so we have majorly cited our own previous results. The discussion part of the current study focuses on the limitations of our previous study and the novel findings of this study (line 291~295, 301~302, 317~320, 327~330).

Reviewer 2 Report

The manuscript by Sun et al. describes the comparison of four promoters: human and porcine elongation Factor 1 alpha (hEF1a, pEF1a), cytomegalovirus (CMV) and the CMV enhancer/chicken beta-actin (CAG) promoter. The promoters drive expression of either a luciferase or HLA-E transgene and were tested in porcine cells. The goal is the identification of promoters for high and ubiquitous expression of xenotransplantation relevant genes in pigs.

Their novel claim is that they identified and characterised the porcine EF1a promoter. Furthermore the pEF1a promoter outperforms the CAG promoter in some cell types, that upon challenge with primate serum its activity is upregulated and that it is not downregulated during prolonged culture.

The paper provides no in vivo data, which would be of interest as the human EF1a promoter functions well in vitro but less so in vivo.

There are some questions which should be addressed prior to any publication:

  • porcine genes are written with capital letters
  • 1: “Green lines indicate regions as promoters selected for use in the current study” the meaning of this sentence is quite clear, please correct.
  • 2 and line 170 it should be mentioned how the “relative luciferase expression levels” were determined.
  • 3A: the Mock vector contains the CAG promoter followed by HLA-E, why does this vector not express HLA-E mRNA?
  • Fig 3B and C: Why is there a discrepancy between the mRNA and protein levels. In PEFs both the mock and CAG constructs show the same level of protein but very different amounts of mRNA. In PK15 cells the CAG construct has the highest mRNA level but the protein level seems similar to mock and lower than for the pEF1a construct?
  • 4 Again there is a discrepancy between the mRNA and protein levels. Also the CAG promoter seems downregulated when treated with primate serum. Is this correct? I am not aware that this has been previously observed.
  • How did the authors exclude that expression differences are not based on different transfection efficiency for the HLA-E constructs?
  • 5 shows that the CAG promoter is downregulated during prolonged culture. Please clarify if day 4 is four days after transfection while the cells are still under Neo selection? If so, than day 4 would still measure transient expression which would be higher than the stable expression after selection and the conclusion would not be correct.

In summary: without answers to the queries stated above, it is difficult to determine if the conclusions drawn by the authors are supported by the results.

Author Response

  • The manuscript by Sun et al. describes the comparison of four promoters: human and porcine elongation Factor 1 alpha (hEF1a, pEF1a), cytomegalovirus (CMV) and the CMV enhancer/chicken beta-actin (CAG) promoter. The promoters drive expression of either a luciferase or HLA-E transgene and were tested in porcine cells. The goal is the identification of promoters for high and ubiquitous expression of xenotransplantation relevant genes in pigs.

Their novel claim is that they identified and characterised the porcine EF1a promoter. Furthermore the pEF1a promoter outperforms the CAG promoter in some cell types, that upon challenge with primate serum its activity is upregulated and that it is not downregulated during prolonged culture.

The paper provides no in vivo data, which would be of interest as the human EF1a promoter functions well in vitro but less so in vivo

There are some questions which should be addressed prior to any publication:

→ We would like to thank the reviewer for careful and thorough reading of this manuscript and for the constructive suggestions, which help to improve the quality of this manuscript. We agree with the reviewer that appending in vivo data will make this study more consistent and convictive. However, it is a time consumptive work and hard to accomplish in short terms. We will disclose more precise and interesting findings in the next phase study.  We have addressed all issues indicated in the review report, and we hope you are satisfied with our revision.

  • porcine genes are written with capital letters

→ The correction has been made (line 229, 232, 240, 248, 254, 264). 

  • 1: “Green lines indicate regions as promoters selected for use in the current study” the meaning of this sentence is quite clear, please correct.

→ Thank you very much for your opinion. The correction has been made (line 237). 

  • 2 and line 170 it should be mentioned how the “relative luciferase expression levels” were determined.

→ Please refer to "SECTION 2.3. Luciferase assay" for the determination method. To avoid confusion, we have revised related sentences (line 123, line 173~174).

  • 3A: the Mock vector contains the CAG promoter followed by HLA-E, why does this vector not express HLA-E mRNA? Fig 3B and C: Why is there a discrepancy between the mRNA and protein levels. In PEFs both the mock and CAG constructs show the same level of protein but very different amounts of mRNA. In PK15 cells the CAG construct has the highest mRNA level but the protein level seems similar to mock and lower than for the pEF1a construct?

→ We used the Mock vector containing the CAG promoter, however, there is no PolyA tail sequence at its 3' end. Therefore, it didn't express HLA-E mRNA.

→ In addition. to improve the readability, the Mock group was removed from Fig. 3 in the revised manuscript (line 248, 254). Because the purpose of this study is to compare the activity of the pEF1α promoter with that of the CAG promoter in different types of porcine cells, we think the Mock group is no longer necessary for the current study and Mock group related data were also exclude from Fig. 4 in the previous version. According to the newly revised Fig. 3B and C, the expression patterns of protein and mRNA are similar, except in PK-15 cells. We speculate that the difference in their expression levels stems from transient transfection (line 248).

  • 4 Again there is a discrepancy between the mRNA and protein levels. Also the CAG promoter seems downregulated when treated with primate serum. Is this correct? I am not aware that this has been previously observed.

→  We did not compare the changes in CAG promoter activity before and after treatment with primate serum. In Fig. 4A and B, the values of CAG are all normalized to "1". The reason why the bar of CAG in Fig. 4B looks more shorter is because the Y-axis scale is different. Therefore, the expression patterns of protein and mRNA in Fig. 4 are consistent (line 254).

  • How did the authors exclude that expression differences are not based on different transfection efficiency for the HLA-E constructs?

→ Thank you very much for the thoughtful comments. In this study, co-transfection controls were designed to remove variability due to transfection efficiency. This point was appended in SECTION 2.2 (line 105~106).

  • 5 shows that the CAG promoter is downregulated during prolonged culture. Please clarify if day 4 is four days after transfection while the cells are still under Neo selection? If so, than day 4 would still measure transient expression which would be higher than the stable expression after selection and the conclusion would not be correct.

→ Day 4 indicates the time after 10 days of Neomycin selection, which is mentioned in SECTION 2.2 (line 108~109).